# Development of Microrobot with Optical Magnetic Dual Control for Regulation of Gut Microbiota

**DOI:** 10.3390/mi14122252

**Published:** 2023-12-17

**Authors:** Xiaotian Lan, Yijie Du, Fei Liu, Gongxin Li

**Affiliations:** The Key Laboratory of Advanced Process Control for Light Industry (Ministry of Education), Institute of Automation, Jiangnan University, Wuxi 214122, China; 6211905023@stu.jiangnan.edu.cn (X.L.); 6231905024@stu.jiangnan.edu.cn (Y.D.); fliu@jiangnan.edu.cn (F.L.)

**Keywords:** *S. cerevisiae*, microrobots, optical field driver, magnetic field driver, regulation of gut microbiota

## Abstract

Microrobots have emerged as a promising precision therapy approach that has been widely used in minimally invasive treatments, targeted drug delivery, and wound cleansing, and they also offer a potential new method for actively modulating gut microbiota. Here, a double-faced microrobot was designed to carry gut bacteria via covalently immobilizing the antibodies, and a corresponding integrated optical and magnetic dual-driving control system was also developed for precise control of the microrobot. The microrobot utilizes magnetic microsphere as its core, with one side coated in gold, which serves as the optical receptor surface and the interface for bacterial attachment. The specific gut bacterium, *S. cerevisiae*, was immobilized on the gold-coated side using the corresponding antibodies. The dual-driving control system enables the precise modulation of gut bacteria by synergistically manipulating the microrobots’ movement via the optical field and magnetic field. The feasibility of independent and coordinated control using optical fields and magnetic fields was validated through experimental and numerical simulation approaches. This work introduces a novel method for the precise modulation of gut microbiota, providing a new avenue for disease treatments based on gut bacteria.

## 1. Introduction

The gut microbiota refers to the community of microorganisms that reside in the intestinal tract and have a symbiotic relationship with the host [1,2]. They play crucial roles in various physiological functions, including energy absorption from food, development and maturation of the immune system, and protection against infections [3,4]. Recently, many reports have highlighted the close association between imbalances in the gut microbiota and several serious diseases, such as cancer, Parkinson’s disease, Alzheimer’s disease, and central nervous system disorders [5,6,7,8]. In clinical practice, the modulation of gut bacteria is typically achieved through dietary interventions and the administration of drugs targeted at the intestinal mucosa [9,10,11]. However, these methods often have low efficiency and face challenges in achieving precise and targeted modulation [12,13,14].

Fortunately, microrobots have emerged as a promising approach for the precise and rapid modulation of gut bacteria. Microrobots have gained attention for their potential applications in disease diagnosis, drug therapy, and even cellular-level surgeries [15,16,17]. By delivering therapeutic substances to specific locations within the body, microrobots offer a new non-invasive and personalized treatment approach [18,19]. However, the small size of microrobots presents challenges in achieving controlled motion, which has become a focal point in the field [20,21,22]. Various novel driving methods have been proposed and implemented [23], including chemical reactions [24], acoustic waves [25], optical radiation [26,27], heat [28], and magnetic fields [29], to provide power and propulsion to microrobots. Among these methods, magnetic and optical field-driven systems have been widely employed due to their versatility, precise control capabilities, and reliability [29,30,31]. Magnetic field drive is widely used in the biomedical field because it can be precisely controlled and is suitable for remote operation and non-destructive to tissues. By introducing magnetic nanoparticles into micro- and nanorobots, magnetic driving force can be obtained simply and effectively. Commonly used magnetic field driving methods include a gradient magnetic field [32], rotating magnetic field [33], and oscillating magnetic field [34]. By designing and manufacturing micro/nanorobots with different structures [35], such as spiral structures, Janus microspheres, etc., faster movement speeds, higher control accuracy, flexible manipulation, and cluster configuration changes can be achieved. Light field drive uses light energy as a kind of wireless energy and has many advantages, such as controllable direction and variable power. Light-driven micro/nanorobots have the characteristics of remote controllability and rapid energy propagation. Light-driven micro/nanorobots are mainly based on photocatalytic reactions, which are driven by generating material or charge gradients near the robot, or using bubbles. Commonly used light-driven materials include photocatalytic materials and photothermal materials [36]. According to different light wavelengths, a suitable driving method can be selected to drive micro/nanorobots. Therefore, both magnetic field drive and light field drive have their own advantages and application scenarios in the field of micro/nanorobots [37]. With the continuous advancement and innovation of technology, these two driving methods will bring more possibilities to the development of micro/nanorobots.

In this study, we designed a double-faced microrobot capable of carrying gut bacteria and developed a corresponding dual-driving control system with optical field and magnetic field. This design enables the microrobot to navigate to specific locations and achieves precise modulation of gut bacteria. The microrobot is a spherical structure with a diameter of 3–7 µm, coated with a 50 nm-thick layer of gold on one side, serving as both a surface for microbial attachment and a contact surface for optical driving [38,39]. Microbial cells are covalently bonded to the gold-coated side. The dual-driving control system consists of magnetic driving structures and optical driving structures, enabling the precise control of microrobot motion to position it at specific locations and adjust microbial concentrations. The magnetic material within the microrobot allows it to be propelled by a magnetic field, while the differential structural properties on the two sides are propelled through self-heating and thermal convection induced by optical driving structure. The dual-driving approach of optical and magnetic forces significantly enhances the accuracy and efficiency of microrobot manipulation, providing a promising avenue for the precise modulation of gut bacteria.

## 2. Materials and Methods

### 2.1. Materials and Instruments

Polystyrene magnetic microspheres containing 20% Fe_3_O_4_ and 70% polystyrene with a diameter of 3–7 µm (Ruige Technology Enterprise Store, Wuxi, China) were chosen as the main body of the microrobots. Gold particles (99.99% purity, Fuzhou Infineon Optoelectronics Technology, Fuzhou, China) were used as the surface gold plating material for the magnetic microspheres. Then, 95% ethanol solution was used to perform surface hydrophilicity treatment on the glass slide. The magnetic field in the experiment was generated by a 3-axis Helmholtz coil, where the coil base and working area were made of white resin material. The current signal in the Helmholtz coil was generated through a power amplifier (OPA548, purchased from Conway Technology, Chengdu, China), which was driven by a signal generator (AFG3152C, purchased from Tec China, Chengdu, China). The near-infrared light used in the experiment was emitted by an externally modulated point laser with a wavelength of 808 nm and a maximum output power of 3200 mW. In addition, a microscope camera (PixeLink, Ottawa, ON, Canada) was also used for the real-time observation. Functionalized microrobots were immobilized with Saccharomyces cerevisiae (*S. cerevisiae*) antibodies using 3-mercaptopropionic acid (3-MPA), 1-ethyl-3-(3-dimethylaminopropyl) carbodiimide hydrochloride (EDC), bovine albumin (BSA), and *N*-hydroxysuccinimide (NHS) as linking agents. Phosphate-buffered saline (PBS) was used to wash the sample. A bicinchoninic acid (BCA) protein detection kit was used for detecting antibodies (purchased from Shanghai Titanic Co., Ltd., Wuxi, China).

### 2.2. Fabrication of the Microrobots

The fabrication process of the microrobots (Figure 1a) is described as follows. A glass slide was chosen as the substrate and subjected to a surface hydrophilicity treatment. First, 1 mL of magnetic microsphere solution with a concentration of 0.17 wt% was dropped onto the glass slide. The microspheres were densely spread on the surface of the glass slide after the microsphere solution dried naturally. Then, a layer of gold with a thickness of 30–50 nm was deposited on the surface of the microspheres by an ion sputtering instrument to form the microrobots, where two half-surfaces were made of different materials. Finally, ultrasound was performed in a pure aqueous solution to collect the prepared microrobots. The structures of the microsphere and microrobot were measured by scanning electron microscope (SEM) (Figure 1b,c).

### 2.3. Structure of the Optical Magnetic Dual-Control System

The optical magnetic dual-control system consists of an optical control part, a magnetic control part, and a microscope, as shown in Figure 2. The optical control part includes a near-infrared light source with a wavelength of 808 nm, an optical power regulator, and a three-dimensional mobile device. The optical power regulator is used to adjust the light intensity, and the three-dimensional mobile device is used to adjust the position of the light illuminated on the microrobot. The magnetic control part consists of a 3-axis Helmholtz coil, three power amplifiers, and a signal generator with three-channels. The 3-axis Helmholtz coil consists of three pairs of mutually orthogonal Helmholtz coils. The signal generator generates a low-frequency sinusoidal current with an amplitude of 50 mV, which is amplified to 1.5 V by the power amplifier, and then is input into the corresponding Helmholtz coil pair. There is a certain phase difference between the sinusoidal signals input to the three pairs of Helmholtz coil to produce a rotating magnetic field, and the direction and strength of the magnetic field can be adjusted by changing the phase difference, frequency, and amplitude of the sinusoidal current. The microrobot is actuated by the rotating magnetic field, and the motion is real-time observed and recorded by the microscope.

### 2.4. Functionalized Microrobots

*S. cerevisiae* was chosen as a typical intestinal bacterium for the application of the microrobots. The microrobot was functionalized by immobilizing *S. cerevisiae* antibodies that specifically bound to the *S. cerevisiae* cells. The specific steps for functionalizing the microrobots are as follows:Add 2 mL of microrobot solution with a concentration of 0.17% to a dish and let it air-dry for 1 h;Add 2 mL of 1 mmol/L 3-MPA solution to the dish and incubate it at room temperature for 4 h, then rinse off the excess solution with deionized water and let it air-dry for 1 h;Add 2 mL of 5 mmol/L EDC/NHS mixed solution to the culture dish and incubate it at room temperature for 3 h, then rinse off the excess solution with deionized water and let it air-dry for 1 h;Add 1 mL of 100 μg/mL *S. cerevisiae* antibodies solution to the dish and incubate it at room temperature for 40 min, then wash off the excess antibodies with 0.1 mM PBS and let it air-dry for 1 h;Add 1% BSA solution to the dish to cover the microrobots and incubate the dish at 4 °C for 30 min, then wash the microrobots with 0.1 mM PBS and let it air-dry;Add a concentration of 100 μg/mL antibody solution to the dish and incubate it at 4 °C for 30 min, then wash the microrobots with 0.1 mM PBS to remove any residue solution. The modification process is now complete.

## 3. Results and Discussion

### 3.1. Magnetic Field Drive System Verification

The 3-axis Helmholtz coil system consists of three pairs of orthogonally arranged Helmholtz coil, which is a platform commonly used to produce a uniform magnetic field, the field strength of which is proportional to the current flowing in the coil. An arbitrary spatial magnetic field can be generated in the working area by the linear combination of the field from these pairs of coils based on the superposition theory. The performance of the 3-axis Helmholtz coil system was analyzed by numerical simulation via a COMSOL(Version:5.6) software, and the simulation results are shown in Figure 3. 1 A current was applied to each coil pair, and a uniform magnetic field was generated at the working area, as shown Figure 3a,b, and the field strength showed an extreme drop to 0 at the boundary of the working area (Figure 3c). In addition, the same strength of the independent magnetic field of the three coil pairs was generated when the equal currents were applied to these coil pairs, as shown in Figure 3d, and the superimposed magnetic flux was 12 T when the input currents were equal to 1 A. The numerical simulation further determined the characteristics and control mechanism of the Helmholtz coil system, and the geometric structure of the coil was also designed based on the magnetic flux obtained from the simulation.

Then, the manipulation performance of the Helmholtz coil system on the microrobots was experimentally validated. The operation was performed by a rotating magnetic field, which was generated by the superposition of the 3-axis magnetic fields induced by the corresponding 3-axis coil pairs being input as alternating currents with a phase difference. The microrobots located in the working area were magnetized by the rotating magnetic field and generated another magnetic field, a so-called “self-magnetic field”. The microrobots rotate when the directions between the self-magnetic field and the Helmholtz coil system are different. And many rotating microrobots are attracted and gathered together by their self-magnetic fields. The actual performance of the Helmholtz coil system in operating microrobots was experimentally validated. First, we adjusted the work area to being level and placed the Petri dish in the work area. Then, we dropped the solution containing the microrobot (0.005 mg/L) into the Petri dish until the solution covered the bottom and the thickness was more than 2 mm. Next, we adjusted the microscope to allow the microrobot to be clearly displayed on the observation screen. The movement trajectory was preset, the appropriate current was selected, and finally, three alternating currents, expressed as $I_{x}=\sqrt{2}\sin4\pi t$, $I_{y}=\sqrt{2}\sin(4\pi t+\pi /2)$, and $I_{z}=0$ were input into these three pairs of coils, respectively, and the microrobots were slowly gathered into a group within 20 s, as shown in Figure 4a. Then, the currents were set as $I_{x}=\sqrt{2}\sin(4\pi t-\pi /6)$, $I_{y}=\sqrt{2}\sin(4\pi t+\pi /6)$, and $I_{z}=\sqrt{2}\sin4\pi t$, and the microrobots were directed to move to the given location, as shown in Figure 4b. The results validate that the Helmholtz coil system can gather the microrobots into a group and manipulate the microrobots at the designed direction.

Through a simulation, it was found that the magnetic field control system built in this study can obtain the required magnetic field by adjusting the size and phase difference of the input current. And after the experimental verification, it was determined that the microrobot produced in this study can reach the preset position under the action of the magnetic field control system, and the function of the magnetic field driving the microrobot to gather and move can be correctly realized.

### 3.2. Optical Control System Verification

Next, the performance of manipulating the microrobots by optical field driving was also validated through an actual experiment. Traditional light control methods mostly adopt bubble driving and self-electrophoretic driving [40], but these methods have significant limitations, such as environmental restrictions, chemical pollution, etc. Here, the microrobots were actuated by the coupling of the self-heating of the microrobots and thermal convection of the liquid after illumination. Due to the inconsistent thermal conductivity coefficient of the two surface materials of the microrobots, the self-heating was generated by the different temperatures of the liquid near the two sides of the microrobots after the microrobots were exposed to light and actuated to move in the liquid in a direction from the Au surface to the other side. Similarly, the thermal convection was generated by natural convection and Marangoni convection between the liquid in the illuminated area and the non-illuminated area. The microrobots move towards the center of convection by the action of liquid convection and attract and gather the adjacent microrobots triggered by static electricity due to a sharp reduction in the gap [41]. The results of the moving microrobots based on these two actuated methods of optical field driving are shown in Figure 5. In Figure 5a, the actuated objects chosen were the polystyrene magnetic microspheres, and the microspheres moved in a straight line to the illuminated area driven only by thermal convection of the liquid. In Figure 5b, the results display that the robots moved towards the illuminated area in a curved motion because the materials on both sides of the microrobots were different, and they not only moved due to the thermal convection of the liquid but also due to their self-heating. Furthermore, Figure 5 also demonstrates that these two coupled driving methods allowed the robot to move faster than by the thermal convection of the liquid alone after the absorption of the same intensity of illumination.

In order to further verify the feasibility and performance of the optical drive part, experiments on the aggregation and movement of the microrobots were conducted, as shown in Figure 6a–e. First, we determined the output power of the light source and adjusted the power density to 1 W/cm^2^. Then, we changed the position of the light spot through the three-dimensional moving platform and illuminated the light spot in the lower left corner of the work area. Then, we turned off the light source and added the solution containing the microrobots (0.005 mg/L). We dropped the solution into the Petri dish until the solution covered the bottom, and the thickness was above 2 mm. Then, we adjusted the microscope so that the microrobot could be clearly displayed on the observation screen. Finally, we turned on the light source and observed the experimental phenomena. The results are shown in Figure 6a. Within 90 s, a large number of microrobots gathered in the illuminated area. Then, we moved the light source in different directions, and the microrobot swarm also moved with the light source, as shown in Figure 6b–e. The results demonstrate that the optical control system is capable of manipulating microrobots to assemble into groups and move in specific directions and trajectories.

Through experimental verification, it was determined that the microrobot produced in this study meets the requirements of light field control. Under the irradiation of near-infrared light with a wavelength of 808 nm, it is simultaneously affected by autothermophoresis and thermal convection. The two work together to make the microrobot gather at the light spot, and it can follow the movement of the light spot and move according to the preset trajectory.

### 3.3. Optical and Magnetic Dual-Drive System Verification

Previous experiments have demonstrated that microrobots can be driven by an optical field and a magnetic field. The optical field-driven approach has higher efficiency for the aggregation and manipulation of microrobots compared to the magnetic field-driven approach, while the magnetic field-driven approach can penetrate some surfaces of objects and apply an actuating force on microrobots [42]. For example, Sun et al. demonstrated the use of thermophoretic forces and phototaxis in microelectromotors, which not only enabled the aggregation of microrobots driven by near-infrared light but also allowed for the formation of various shapes by adjusting the shape of the light spot. In contrast, achieving the aggregation of microrobots into irregular shapes using magnetic fields is considerably more challenging. This example supports the notion that optical field driving is relatively more effective in driving microrobot aggregation compared to magnetic fields. The collaborative control of the optical field-driven and magnetic field-driven allow microrobots to have more flexible movement and applications. So, controlling the microrobots by both field-driven approaches was experimentally validated. First, the microrobots were driven to gather using a light field. We adjusted the power density of the light source to 1 W/cm^2^ and adjusted the three-dimensional mobile platform to modulate the light spot in the middle of the working area. Then, we dripped the microrobot solution and turned on the light source. We adjusted the microscope so that the microrobot could be clearly displayed on the observation screen. After 110 s, the microrobots gathered into a group at the light spot; the experimental results are shown in Figure 7a. Then, the magnetic field-driven method was used to move the microrobots in a given direction, as shown in Figure 7b. Here, three alternating currents, $I_{x}=\sqrt{2}\sin(4\pi t-\pi /6)$, $I_{y}=\sqrt{2}\sin(4\pi t+\pi /6)$ and $I_{z}=\sqrt{2}\sin(4\pi t+\pi )$, were input into these three pairs of coils, respectively. The experiment combined optical and magnetic control to verify the possibility of collaborative work between an optical and magnetic dual-driver.

### 3.4. Drug Loading Verification

Finally, the functionalized microrobots were chosen as the manipulated object to validate the effectiveness of the optical–magnetic dual control method. The gold-deposited side of the microrobot was covalently linked to *S. cerevisiae* antibodies through EDC/NHS compounds, and then the antibody was bound to *S. cerevisiae* specifically to achieve functional characterization of the microrobot, as shown in Figure 8a. The successful functionalized microrobots were validated by the protein concentration detection reagent. The microrobots without any modification of antibodies and functionalized with *S. cerevisiae* antibodies were added to the detection reagent, and the results show that the color of the reagent containing *S. cerevisiae* antibodies turned blue, which indicates that the microrobot successfully immobilized the antibody protein, as shown in Figure 8b. The final verification focused on the ability of the microrobots to transport antibodies to the target area. First, the functionalized microrobot solution was dropped onto a glass slide. Then, the light source density was set to 1 w/cm^2^, and the light spot was irradiated on the upper left side of the pipe. It can be clearly observed that after 90 s, the microrobots gathered in the pipe above, as shown in Figure 8c. Then, three alternating currents, $I_{x}=\sqrt{2}\sin(4\pi t-\pi /6)$, $I_{y}=\sqrt{2}\sin(4\pi t+\pi /6)$ and $I_{z}=\sqrt{2}\sin(4\pi t+\pi )$, were input into these three pairs of coils. The microrobots passed through the channel within 50 s, as shown in Figure 8d. The experiment successfully mounted *S. cerevisiae* cells on a microrobot, verifying the feasibility of the concept of microrobots carrying cells. And on this basis, the microrobots that were equipped with cells were driven to gather and successfully passed through the simulated blood vessels, verifying the study’s concept of drug transportation and realizing the function of specifically adjusting the concentration of *S. cerevisiae*.

## 4. Conclusions

A double-faced microrobot that is capable of carrying gut microbiota was designed, and a corresponding dual-driving control system combining optics and magnetism was also developed, enabling the transport of the microrobot carrying gut bacteria to specific locations and achieving precise regulation. A magnetic microsphere coated with Au on one side was used as the core structure of the microrobot and served as a surface for microbial attachment. Coating the microsphere with an Au layer on one side allowed the microrobot to be controlled by the dual-driving mechanism of optics and magnetism. For magnetic driving, a 3-axis Helmholtz coil magnetic field was employed to generate a rotating magnetic field that controlled the microrobot’s controllable motion in three-dimensional space. In terms of optical driving, the near-infrared light was chosen as the driving source, and the microrobots were controlled to carry out rapid aggregation and movement by the combined driving force of self-heating and thermal convection in the solution. Finally, the feasibility was validated for the rapid regulation of gut microbiota by covalently linking *S. cerevisiae* cells to the surface of the microrobot and performing targeted delivery using the control system. In summary, this study provides a novel approach for the precise regulation of gut microbiota.

## Figures and Tables

**Figure 1 micromachines-14-02252-f001:**
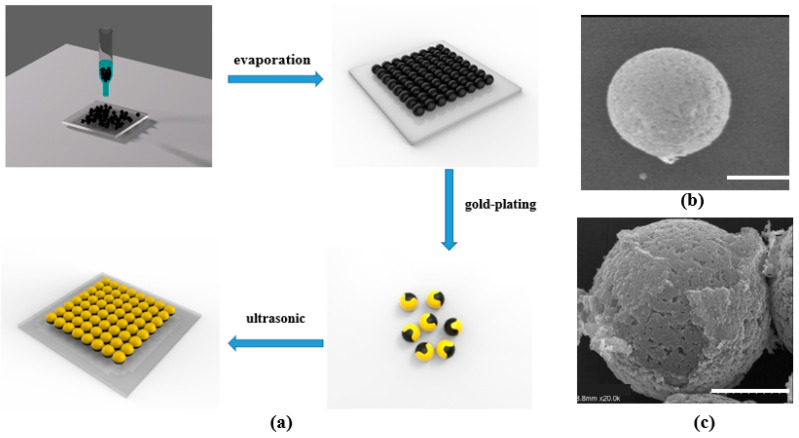
The production process and the characterize of the microrobot. (**a**) The production process of microrobots. (**b**) SEM image of the magnetic microsphere. (**c**) SEM image of the microrobot. Bar scale: 2.0 μm.

**Figure 2 micromachines-14-02252-f002:**
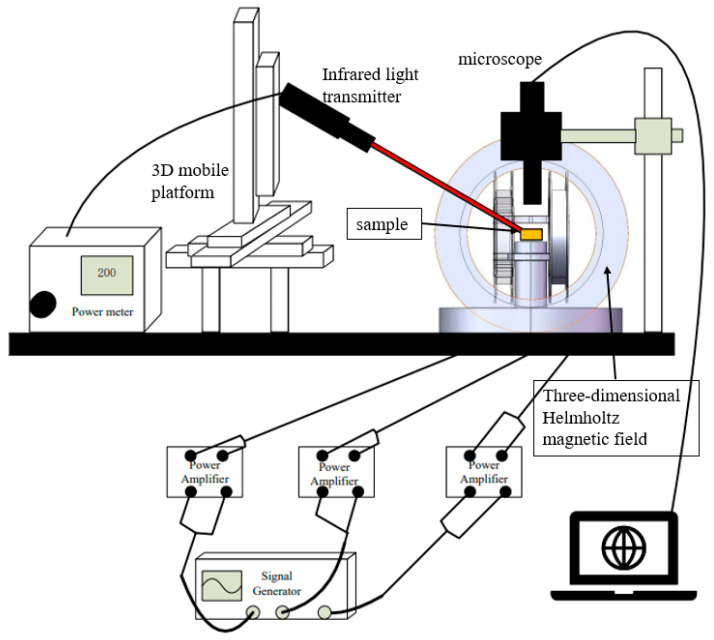
Construction of optical magnetic dual-control system.

**Figure 3 micromachines-14-02252-f003:**
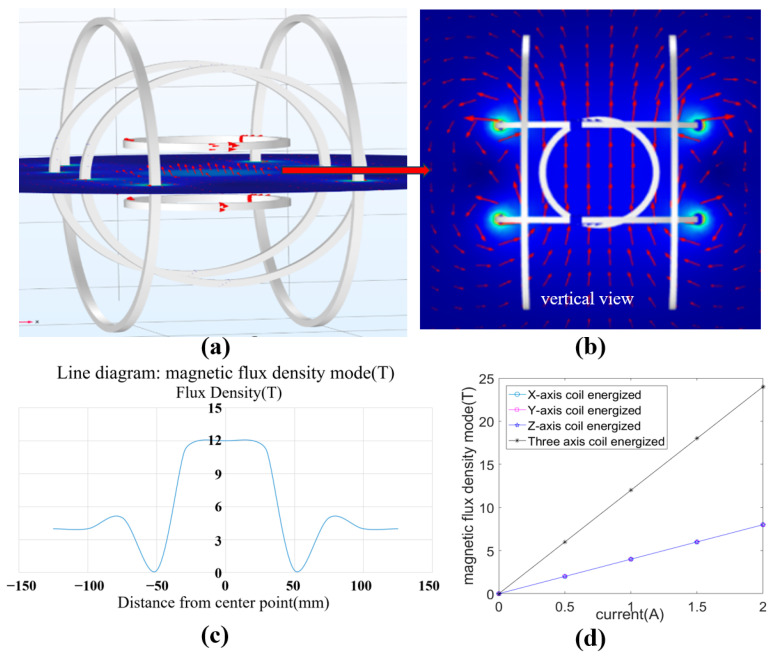
Numerical simulation of the 3-axis Helmholtz coil magnetic field. (**a**) The magnetic distribution of the 3-axis Helmholtz coil applied 1 A currents. (**b**) The magnetic distribution from the vertical view (The field of view is from the top view of **a**). (**c**) Magnetic flux density of the working area. (**d**) The variation in the magnetic induction intensity with different input currents.

**Figure 4 micromachines-14-02252-f004:**
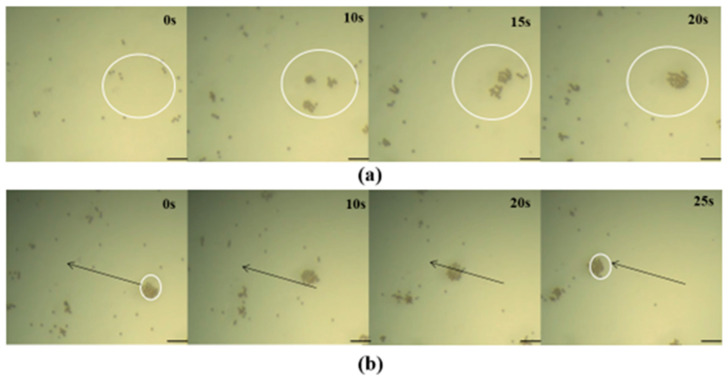
The experimental results of the magnetic field-driving. (**a**) Microrobot convergence process. (**b**) Microrobots moving along in the given direction. Bar scale: 50 μm.

**Figure 5 micromachines-14-02252-f005:**
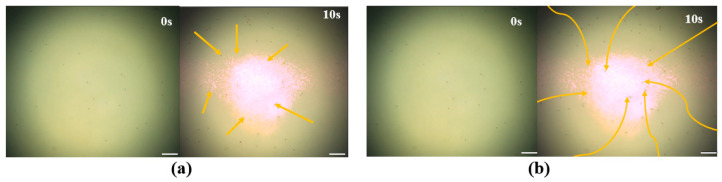
The manipulation results of the optical field-driving experiment, The yellow line indicates the movement trajectory of the microrobot. (**a**) The movement trajectory of the microsphere. (**b**) The movement trajectory of the microrobots. Scale bar: 50 μm.

**Figure 6 micromachines-14-02252-f006:**
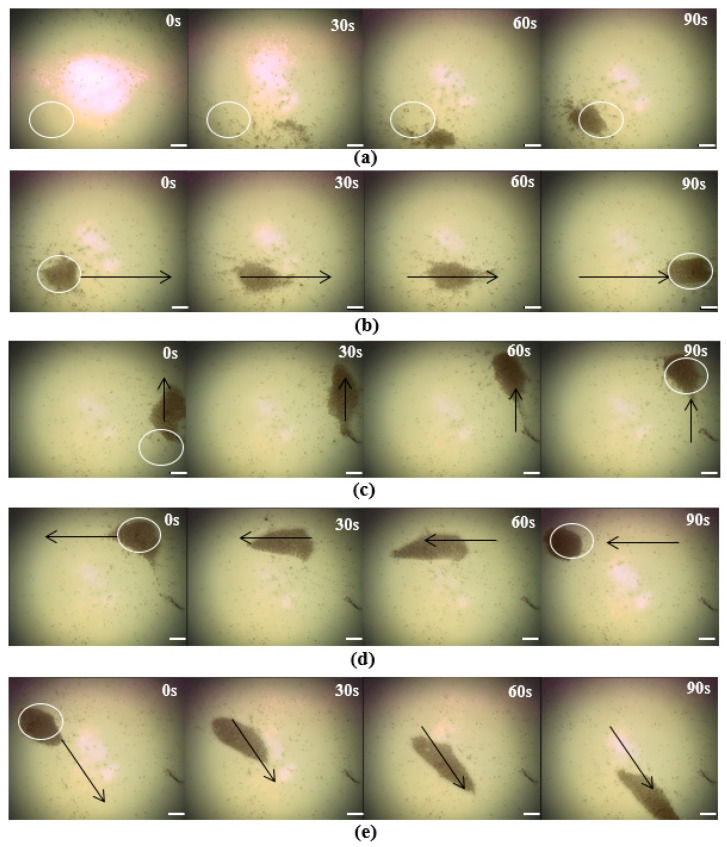
The results of manipulating the microrobots using an optical field. (**a**) The microrobots converged in the lower left corner using the optical field. (**b**–**e**) The microrobots moved in the direction of light movement. The black arrow indicates the direction of light movement. Scale bar: 50 μm.

**Figure 7 micromachines-14-02252-f007:**
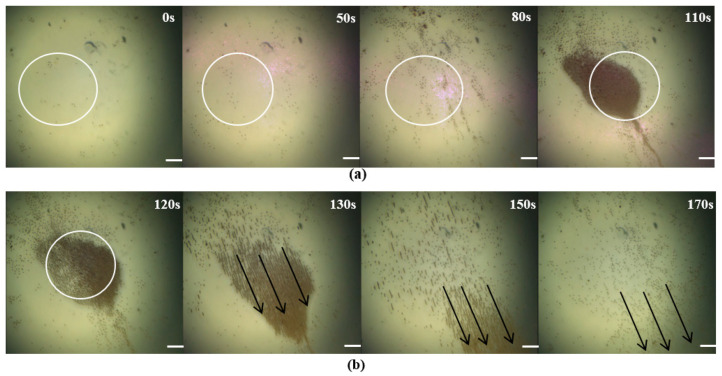
The manipulation results of the optical field and magnetic field together, The white circle indicates the location of the convergence point, and the black arrow indicates the movement direction of the microrobot. (**a**) The rapid aggregation process of microrobots using optical field-driving. (**b**) The microrobots moved in the given direction using the magnetic field. Scale bar: 50 μm.

**Figure 8 micromachines-14-02252-f008:**
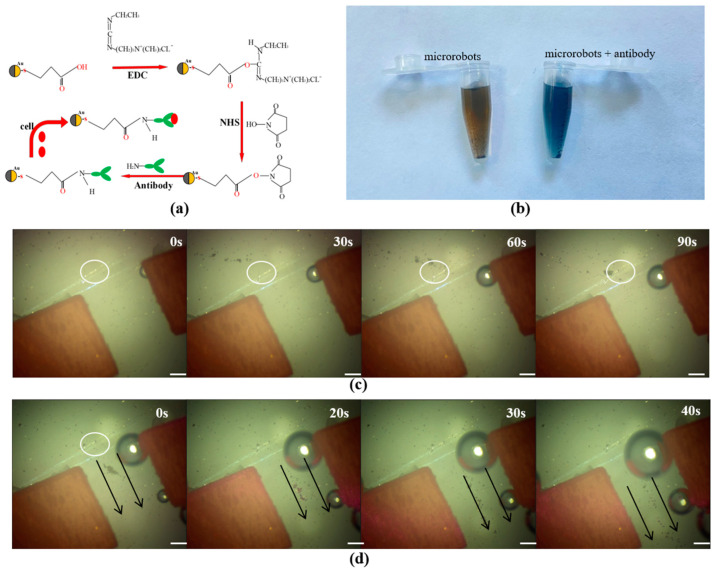
Manipulation of the functionalized microrobots at the designed channel, The white circle indicates the location of the convergence point, and the black arrow indicates the movement direction of the microrobot. (**a**) Schematic diagram of functionalized microrobots. (**b**) Test for microrobots by the protein detection reagent. (**c**) The microrobots gathered by the optical field-driver. (**d**) The microrobots moved by the magnetic field-driver at the given channel. Scale bar: 50 μm.

## Data Availability

Data are contained within the article.

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
