# Peer review of "Development of Microrobot with Optical Magnetic Dual Control for Regulation of Gut Microbiota"

_micromachines, 2023, doi:10.3390/mi14122252_

Round 1

Reviewer 1 Report

Comments and Suggestions for Authors

This study introduces a dual-faced microrobot with an integrated optical-magnetic control system for targeted transport and precise modulation of gut microbiota. The microrobot, featuring a gold-coated magnetic microsphere, responds to optical and magnetic driving mechanisms. Magnetic fields enable controlled motion, while optical fields induce rapid aggregation. Covalently linked S. cerevisiae cells validate the microrobot's capability for targeted delivery. Overall, this innovative approach offers precise regulation of gut microbiota, presenting a promising avenue for therapeutic applications.

The manuscript is well-written and generally clear. In the vast majority of cases, the experimental content can support its conclusions. The paper shows substantial promise, despite some limitations and areas for improvement identified.

Technical Soundness:

(1). First of all, sorry for my limited background knowledge in the field of material processing. I noticed that the authors mentioned the diameter of Polystyrene magnetic microspheres is 3-7 micrometers (page 2, lines 71-72). After the processing in Section 2.2, did the size of microrobots reach around 200 micrometers (Figure 1b and c)?  I can't see the numbers on the scale in the picture clearly, but it looks like it's around 200 micrometers. Is the size of the obtained microrobots stable? It would be preferable to provide a microscopic image, showing several or dozens of particles, to allow readers to visually assess the stability of the microrobots' fabrication using this method.

(2). In Figure 6b to e, the authors only provide the positions of the light at 0s and 90s. Could you clarify whether the light moves in a uniform or non-uniform manner? If it is moving uniformly, could you provide information on how the authors selected this speed? Additionally, are the microrobots consistently moving along with the light source (at the same or similar speed)? Please provide an accurate description of your procedures and the observed experimental phenomena.

(3). Page 6 line 221-224. Why do the authors assert that the optical field-driven approach is more effective in aggregating and manipulating microrobots compared to the magnetic field-driven approach, while the magnetic field-driven approach can penetrate the surface of some objects and apply an actuating force on the microrobot? I haven't found any experimental comparisons in the preceding content to support this conclusion. Or, is there any reference that validates this assertion?

(4). Page 6 line 227-228. The authors state in the main text that the microrobots in Figure 7a are driven by a magnetic field, but in the caption of Figure 7a, the authors mention that they are driven by the optical field. Please verify.

(5). Are the two bright spots in Figure 8 bubbles, and do they seem to be gradually increasing in size? Please provide a brief explanation.

Minor problems:

(1). The full name should be provided when using SEM for the first time. Please review the entire paper to verify if there are similar issues with other abbreviations.

(2). Please provide the Bar scale value in the caption of Figure 8.

Comments on the Quality of English Language

Grammar:

The main text and figures captions contain many grammar errors. Although they do not hinder comprehension, correcting these errors is necessary for a scholarly paper. Below, I have listed several grammar errors from the first four pages. Please carefully review the entire manuscript and correct all grammar errors.

(1). Page 2 line 53, ‘This design enables the microrobot to navigate to specific locations and achieve precise modulation of gut bacteria.’ The word 'achieve' should be 'achieves'.

(2). Page 2 line 74, The word 'is' should be 'are'.

(3). Page 3 line 96, Should the word 'which' be changed to 'where'?

(4). Page 3 line 97, Should the last 'the' be deleted?

(5). Page 4 line 162, Should the word 'generates' be changed to 'generate'?

Reviewer 2 Report

Comments and Suggestions for Authors

The authors developed an integrated optical and magnetic dual-driving control system to manipulate the bacteria. Their results support the claims. There are some deficiencies that need to be corrected as listed below: 

1.     Will the self-heating and thermal convection induced by the optical driving influence the activity of bacteria. What is the laser power used in the experiments.

2.     Why did the authors select the 808 nm wavelength?

3.     Will EDC/NHS solution influence the activity of bacteria? The bacterial activity after conjugation could be analyzed.

4.     Sentences in line 220 need more references.

5.     What is the manipulation resolution of the optical driving system and the magnetic driving system?
